# Structural insights into sodium transport by the oxaloacetate decarboxylase sodium pump

Xin Xu[1†], Huigang Shi[2,3†], Xiaowen Gong[4‡], Pu Chen[1], Ying Gao[4], Xinzheng Zhang[2,3]*, Song Xiang[1]*

[1]Department of Biochemistry and Molecular Biology, Key Laboratory of Immune Microenvironment and Disease (Ministry of Education), Tianjin Medical University, Tianjin, China; [2]National Laboratory of Biomacromolecules, CAS Center for Excellence in Biomacromolecules, Institute of Biophysics, Chinese Academy of Sciences, Beijing, China; [3]University of Chinese Academy of Sciences, Beijing, China; [4]CAS Key Laboratory of Nutrition, Metabolism and Food safety, Shanghai Institute of Nutrition and Health, Shanghai Institutes for Biological Sciences, Chinese Academy of Sciences, Shanghai, China

**\*For correspondence:**
xzzhang@ibp.ac.cn (XZ);
xiangsong@tmu.edu.cn (SX)

[†]These authors contributed equally to this work

**Present address:** [‡]Bio-X Institutes, Key Laboratory for the Genetics of Developmental and Neuropsychiatric Disorders (Ministry of Education), Shanghai Jiao Tong University, Shanghai, China

**Competing interests:** The authors declare that no competing interests exist.

**Abstract** The oxaloacetate decarboxylase sodium pump (OAD) is a unique primary-active transporter that utilizes the free energy derived from oxaloacetate decarboxylation for sodium transport across the cell membrane. It is composed of 3 subunits: the α subunit catalyzes carboxyl-transfer from oxaloacetate to biotin, the membrane integrated β subunit catalyzes the subsequent carboxyl-biotin decarboxylation and the coupled sodium transport, the γ subunit interacts with the α and β subunits and stabilizes the OAD complex. We present here structure of the *Salmonella typhimurium* OAD βγ sub-complex. The structure revealed that the β and γ subunits form a $\beta_{3}\alpha_{3}\beta_{3}\gamma_{3}$ hetero-hexamer with extensive interactions between the subunits and shed light on the OAD holo-enzyme assembly. Structure-guided functional studies provided insights into the sodium binding sites in the β subunit and the coupling between carboxyl-biotin decarboxylation and sodium transport by the OAD β subunit.

## Introduction

The sodium gradient across the cell membrane is utilized in many microorganisms as an energy source to drive ATP synthesis, nutrient intake, cell motility and other processes (*Mulkidjanian et al., 2008*). The oxaloacetate decarboxylase sodium pump (OAD) is the first identified sodium pump responsible for maintaining the sodium gradient in microorganisms (*Dimroth, 1980*). It is found in many bacteria and archaea including the human pathogens *Klebsiella aerogenes* (*Dimroth, 1980*), *Klebsiella pneumonia* (*Schwarz et al., 1988*), *Salmonella typhimurium* (*Wifling and Dimroth, 1989*) and *Legionella pneumophila* (*Jain et al., 1996*). It plays a critical role in the anaerobic citrate fermentation pathway (*Dimroth et al., 2001*), and has been shown to be important for the pathogenicity of certain pathogens (*Hwang et al., 1992*; *Jain et al., 1996*). OAD utilizes the free energy derived from oxaloacetate decarboxylation to drive sodium transport across the cell membrane. It is composed of α, β and γ subunits. Biotin serves as the carrier for the carboxyl group in its reaction and is covalently linked to the biotin-carboxyl carrier protein (BCCP) domain in the α subunit C terminus. The carboxyltransferase (CT) domain in the α subunit catalyzes the carboxyl-transfer from oxaloacetate to biotin; the β subunit catalyzes the subsequent carboxyl-biotin decarboxylation and the coupled sodium transport; the γ subunit stabilizes the OAD complex by interacting with the α and β subunits (*Buckel, 2001*; *Dimroth et al., 2001*).

OAD belongs to the decarboxylase sodium pump family (*Buckel, 2001*; *Dimroth et al., 2001*). Members in this family also include methylmalonyl-CoA decarboxylases (MCD), glutaconyl-CoA decarboxylase (GCD) and malonate decarboxylase (MAD). They utilize the free energy derived from decarboxylating the relevant substrates to drive sodium transport across the cell membrane, and constitute a unique family of primary-active transporters (*Saier et al., 2016*). MCD and GCD catalyze a two-step decarboxylation/sodium transport reaction similar to the OAD reaction (*Buckel, 2001*). Their CT and BCCP domains are encoded by separated α and γ subunits, respectively. Their β subunits are highly homologous to the OAD β subunit (*Figure 1—figure supplement 1a*). They also form stable complexes and contain a scaffolding δ subunit homologous to the OAD γ subunit (*Figure 1—figure supplement 1b*; *Buckel, 2001*). MAD catalyzes a more complex reaction. The *Malonomonas rubra* MAD transfers the carboxyl group in malonate to a specific acyl carrier protein before to biotin. Its MadB component catalyzes the subsequent carboxyl-biotin decarboxylation and sodium transport (*Dimroth and Hilbi, 1997*). MadB is homologous to the OAD β subunit (*Figure 1—figure supplement 1a*).

Structural studies of the CT domains in OAD (*Studer et al., 2007*) and GCD (*Wendt et al., 2003*) have provided insights into their substrate decarboxylation. Homologues of the CT and BCCP domains in decarboxylase sodium pumps are found in many biotin-dependent enzymes. Structural studies on these enzymes have also shed light on their function (*Jitrapakdee and Wallace, 2003*; *Tong, 2013*; *Waldrop et al., 2012*). In contrast, little is known about the structure of other components in decarboxylase sodium pumps, and the mechanism of their sodium transport is poorly understood. We present here structure of the *Salmonella typhimurium* OAD (*St*OAD) βγ sub-complex. The structure revealed that the β and γ subunits form a $\beta_{3x03B3}\gamma_3$ hetero-hexamer with extensive interactions between subunits and provides insights into the OAD holo-enzyme assembly. The structure and structure-guided functional studies provided insights into the sodium binding sites in the β subunit and identified a number of residues critical for its function. Our data suggest an 'elevator mechanism' for the sodium transport by the OAD β subunit and provide insights into its coupling with carboxyl-biotin decarboxylation.

## Results

### Overall structure of the *St*OAD βγ sub-complex

We initially attempted to study the structure of the *St*OAD holoenzyme and expressed it in *Escherichia coli* (*E. coli*) and purified it through a 6x Histidine tag engineered to the N-terminus of the γ sub-unit and nickel-nitrilotriacetic acid (Ni-NTA). The purified complex was crystallized with vapor diffusion, and the crystals diffracted to a maximum resolution of 4.4 Å at the Shanghai Synchrotron Radiation Facility. Sodium dodecyl sulfate (SDS) polyacrylamide gel electrophoresis (PAGE) analysis revealed that they do not contain the α subunit, suggesting that it dissociated during crystallization and the βγ sub-complex is more stable. We further purified the βγ sub-complex, which appears to be well-folded judged by our gel filtration experiments (*Figure 1—figure supplement 2*). Although it crystallized, the crystals did not diffract better. Dynamic light scattering experiments indicated that it has a molecular weight of 250 kDa, suggesting that it is suitable for cryo-electron microscopy (EM) studies. We carried out cryo-EM experiments and determined its structure (*Figure 1—figure supplement 3a–d*) to an overall resolution of 3.88 Å (*Figure 1—figure supplement 3e*). Resolution of the core regions reaches 3.6 Å (*Figure 1—figure supplement 3f*). The excellent density (*Figure 1—figure supplement 4a–e*) allowed us to place all the non-hydrogen atoms for residues 13–432 in the β subunit. Additional poorer densities were found for an α helix about 40 amino acids in length (*Figure 1—figure supplement 4f*). Since most residues in the β subunit were accounted for, residues 2–43 in the γ subunit were assigned to this helix based on secondary structure prediction and density features. The structure agrees well with the experimental data and expected geometric values (*Table 1*). Although 100 mM of sodium chloride was included in the protein buffer, no obvious densities for sodium ions were observed, probably due to the moderate resolution. Using the cryo-EM structure as the search model, we determined the crystal structure of the βγ sub-complex by molecular replacement (*Supplementary file 1A*). No major conformational differences were found between the cryo-EM and the crystal structures. In the remainder of the manuscript we will mainly discuss the cryo-EM structure, since its resolution is higher.

**Table 1.** Cryo-EM data collection and structure refinement statistics

| | Cryo-EM structure of the *St*OAD βγ sub-complex (PDB 6IWW) |
|---|---|
| **Data collection and procession** | |
| Magnification (nominal) | 105,000 |
| Voltage (kV) | 300 |
| Electron exposure (e⁻/Å²) | 50 (32 frames) |
| Defocus range (μm) | 1.0–3.5 |
| Pixel size (Å) | 0.68 |
| Symmetry imposed | C3 |
| Initial particle images (no.) | 260,982 |
| Final particle images (no.) | 75,699 |
| Map resolution (Å) | 3.88 |
| FSC threshold | 0.143 |
| Refinement | |
| Initial model used | EMBUILDER generated model |
| Model resolution (Å) | 3.88 |
| FSC threshold | 0.143 |
| Map sharpening B factor (Å²) | −250 |
| Model composition | |
| Non-hydrogen atoms | 10,227 |
| Protein residues | 1386 |
| Ligands | 3 |
| B-factors (Å²) | |
| Protein | 41.7 |
| Ligand | 23.9 |
| R.m.s deviations | |
| Bond lengths (Å) | 0.008 |
| Bond angles (°) | 1.016 |
| Validation | |
| Molprobity score | 1.61 |
| Clashscore | 5.83 |
| Poor rotamers (%) | 0 |
| Ramachandran plot | |
| Favored | 95.78 |
| Allowed (%) | 4.22 |
| Disallowed (%) | 0 |

The structure revealed that the β and γ subunits form a $β_{3x03B3}γ_3$ hetero-hexamer (*Figure 1a–b*). The expected molecular weight estimated by dynamic light scattering is larger than that of the hexamer (168 kDa), likely due to detergent molecules bound to the hexamer. The β subunits form a trimer at the center of the hexamer, the γ subunits are located in the periphery. One side of the β subunit trimer is predominantly positively charged, the opposite side is negatively charged (*Figure 1c–d*). The 'positive inside rule' (*von Heijne, 1992*) indicates that the positively charged side is located in the cytoplasm. This side is also concentrated with conserved residues (*Figure 1—figure supplement 5a–d*). The structure revealed no solvent accessible channels connecting the cytoplasm and the periplasm.

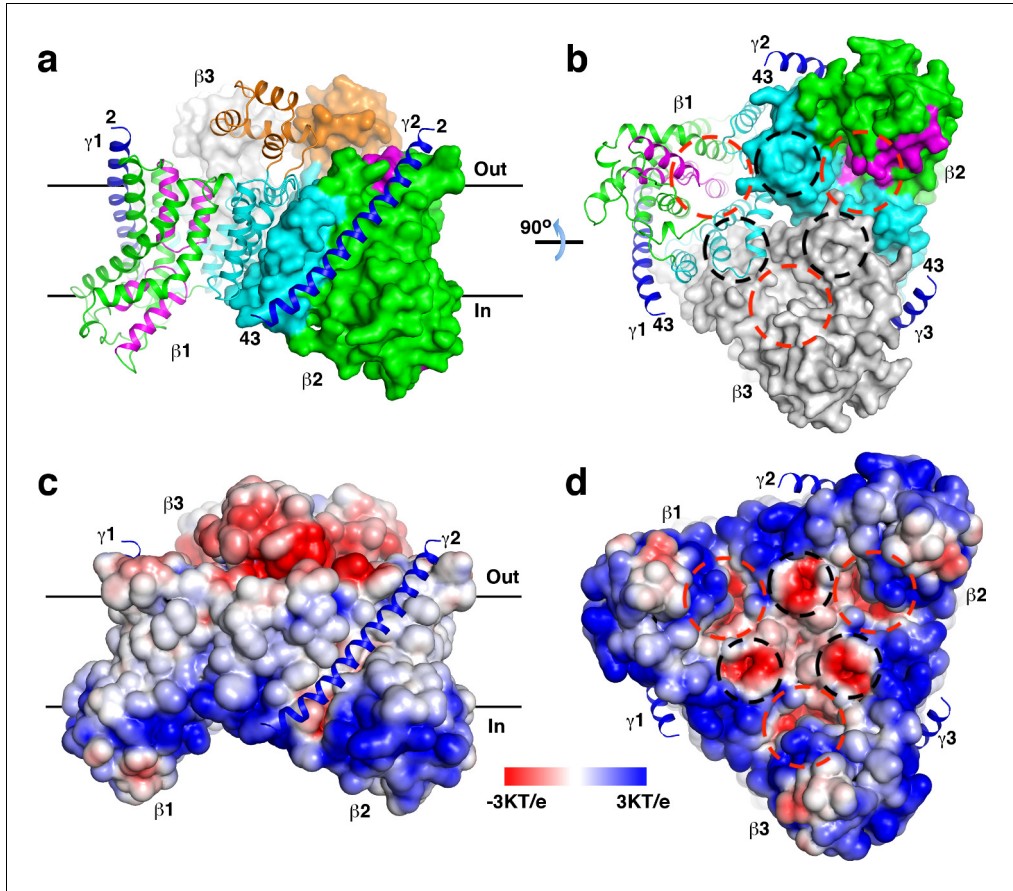

**Figure 1.** Overall structure of the *St*OAD βγ sub-complex. (**a**)-(**b**) Structure of the *St*OAD β$_{3x03B3}$γ$_3$ hetero-hexamer. The γ subunits are colored in blue and shown in cartoon representation. The first and second β subunits (β1 and β2) are colored with the scaffold domain in cyan, core domain in green, domain E in orange and the helical hairpins in magenta. This coloring scheme is used throughout the manuscript unless otherwise indicated. The third β subunit (β3) is colored in gray. Cartoon representation of β1 and the molecular surfaces of β2 and β3 are shown. The first and last residues in the γ subunit visible in our structure are indicated (residues 2 and 43, respectively). Structural figures were prepared with PyMOL (www.pymol.org). (**c**)-(**d**) Electrostatic potential at the solvent assessable surface of the *St*OAD β subunit trimer. The views of panels (**c**) and (**d**) are identical to the views of panels (**a**) and (**b**), respectively. The γ subunits are presented in cartoon representation and colored in blue. The red and black circles in panels (**b**) and (**d**) indicate the first and second negatively charged regions in the β subunit cytoplasmic face, respectively. The black lines in panels (**a**) and (**c**) indicate the boundaries of the membrane bilayer.

The online version of this article includes the following figure supplement(s) for figure 1:

**Figure supplement 1.** Sequence alignment of the OAD β (**a**) and γ (**b**) subunits and their equivalents in other decarboxylase sodium pumps.

**Figure supplement 2.** Gel filtration characterization of the wild type and substituted *St*OAD βγ sub-complex.

**Figure supplement 3.** Structure determination of the *St*OAD βγ sub-complex by cryo-EM.

**Figure supplement 4.** Cryo-EM densities.

**Figure supplement 5.** Conservation of individual residues in the β subunit.

**Figure supplement 6.** The γ subunit C-terminal tail plays a critical role in the interaction between the α subunit and the βγ sub-complex.

---

The structure indicates that the γ subunit C-terminal tail is located in the cytoplasm and could mediate interactions with the α subunit (*Figure 1a–b*). This is in line with previous reports that the isolated γ subunit C-terminal tail interacts with the α subunit (*Balsera et al., 2011*; *Dahinden et al., 2005*), and the α subunit fails to interact with γ subunit variants with mutations or deletions in this region (*Schmid et al., 2002*). To directly test if the γ subunit C-terminal tail mediates interactions

between the α subunit and the βγ sub-complex, we generated a ΔγCT variant of the StOAD expression construct lacking residues 61–84 in the γ subunit. The removed residues span the predicted amphipathic helix (*Figure 1—figure supplement 1b*) in the *Vibrio cholerae* OAD isoform 2 (*Vc*OAD2) that has been proposed to bind the α subunit association domain (AD) to form a structure similar to the pyruvate carboxylase (PC) tetramerization (PT) domain (*Balsera et al., 2011*). We found that the ΔγCT truncation did not affect the protein level of the α subunit in the soluble fraction (*Figure 1—figure supplement 6a*) or the interaction between the β and γ subunits (*Figure 1—figure supplement 6b*). However, it severely reduced the amount of the α subunit co-purified with the βγ sub-complex (*Figure 1—figure supplement 6b*). These data indicate that the γ subunit C-terminal tail is not involved in the interaction between the β and γ subunits but plays a critical role in the interaction between the α subunit and the βγ sub-complex.

## Structure of the StOAD β subunit

The β subunit is composed of 10 transmembrane helices (T1-10), two helical hairpins (H1a/b and H2a/b) and a number of additional helices (*Figure 1a* and *Figure 2a–c*). Its N- and C-terminal halves form two repeats of inverted topology (*Figure 2c*). The region between T2 and T3' contains four helices (αE1-4), which form a solvent exposed domain E. This domain is poorly conserved and does not exist in OAD in some organisms (*Figure 1—figure supplement 1a*). H1 and H2 point into the interior of the β subunit from opposite sides. They and surrounding helices T4-5 and T9-10 form a

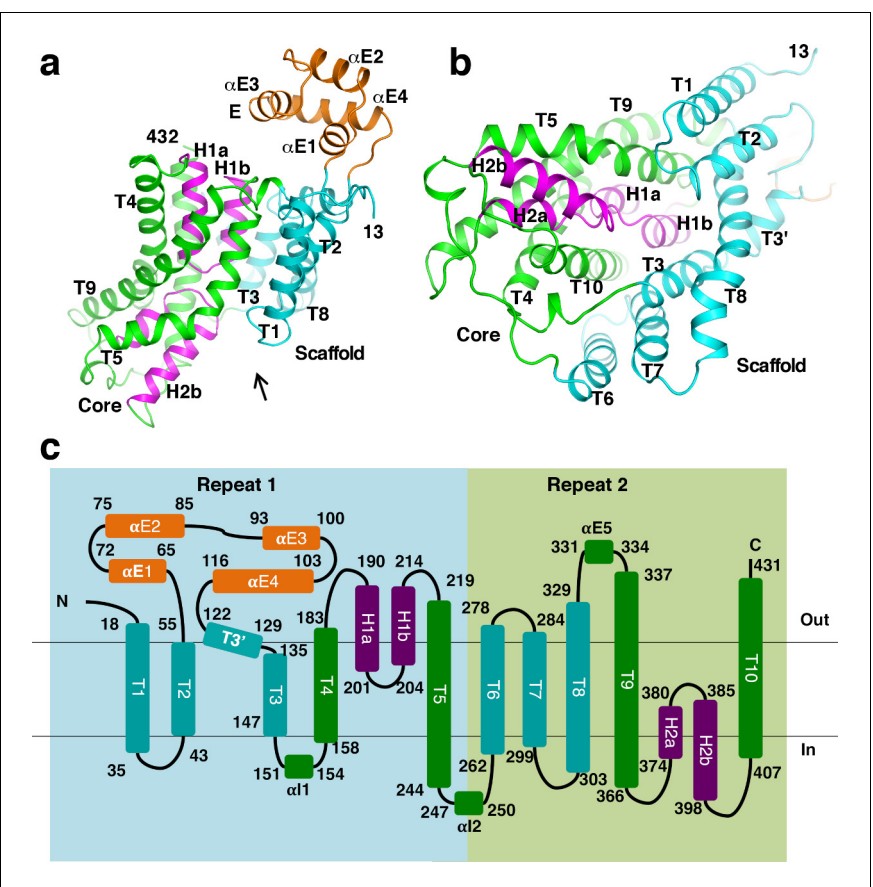

**Figure 2.** Structure of the β subunit. (a)-(b) Structure of the StOAD β subunit. In panel (b), one views along the arrow indicated in panel (a). (c) Topology of the β subunit. Boxes in light blue and light green indicate the inverted repeats in the β subunit.

The online version of this article includes the following figure supplement(s) for figure 2:

**Figure supplement 1.** Structural homology between the StOAD β subunit and CitS.

**Figure supplement 2.** Structural homology between the StOAD β subunit and NapA.

bundle-like core domain. The tips of H1 and H2 meet at the middle of the core domain. The rest of the β subunit forms a scaffold domain, surrounding the core domain at one side (*Figure 2a–b*).

A search through published structures (*Holm and Rosenström, 2010*) indicated that the closest structural homologue of the β subunit is the sodium/citrate symporter CitS (*Kim et al., 2017*; *Wöhlert et al., 2015*). The structures of the *Salmonella enterica* CitS (*Se*CitS) and the β subunit can be superimposed with a root mean square deviation of 4.1 Å for equivalent Cα atoms (*Figure 2—figure supplement 1a–b*), despite their limited sequence identity (8%, *Figure 2—figure supplement 1c*). All major secondary structure elements in the β subunit core and scaffold domains have equivalents in CitS, which contains an extra helix at its N-terminus. The β subunit belongs to the cation:proton antiporter (CPA) superfamily and has been suggested to have homology to proteins in the CPA2 family (http://www.tcdb.org/search/result.php?tc = 3.B.1). Consistently, we found that the *St*OAD β subunit structure is homologous to the structure of the sodium/proton antiporter NapA (*Figure 2—figure supplement 2a–b*; *Coincon et al., 2016*; *Lee et al., 2013*), a member of the CPA2 family. It also has homology to sodium/proton antiporters NhaP (*Paulino et al., 2014*; *Wöhlert et al., 2014*) and NhaA (*Hunte et al., 2005*; *Lee et al., 2014*), the sodium/bile acid symporter ASBT (*Hu et al., 2011*; *Zhou et al., 2014*) and others. The homology is pronounced for the core domain. The equivalent regions in these β subunit homologues contain two discontinuous transmembrane helices in the place of H1 and H2 (*Figure 2—figure supplement 2a*). The scaffold domain also shows significant homology to equivalent regions in NapA and NhaP, which contain an extra helix (*Figure 2—figure supplement 2b*).

## Interactions between subunits

Extensive interactions between the β subunits are observed in our structure. The interactions are primarily mediated by the scaffold domain (*Figure 3a–b*). At the outer part of the trimer interface, T1 and T2 interact with T6 and T7 in the neighboring monomer. Towards the center, interactions are mediated by T3', T3 and T8. These interactions are primarily mediated by hydrophobic residue side chains. 2700 Å$^2$ of surface area is buried at each interface between the scaffold domain in different monomers. Helix αE4 in domain E also makes a minor contribution to the trimer interface. 250 Å$^2$ of surface area is buried at each interface between αE4 in different monomers.

Unlike the β subunit, the homologous CitS forms a dimer (*Kim et al., 2017*; *Wöhlert et al., 2015*). Structural analysis indicated that the large conformational differences between T1 and T6 and their equivalents in CitS (*Figure 2—figure supplement 1b*) plays a key role in determining the oligomerization states. In the β subunit, T1 and T6 are located close to core domain. In CitS, the equivalent H2 and H8 are located away from the core domain (*Figure 3—figure supplement 1a*). H2 and H1 N-terminal to it interact with H8 in the other monomer in the CitS dimer (*Figure 3c* and *Figure 3—figure supplement 1a*). Compared to equivalent regions in the *St*OAD β subunit, these helices create a 'wedge' at the interface between monomers, which promotes dimer formation (*Figure 3c–d*). Additional homologues of the β subunit including NapA (*Coincon et al., 2016*; *Lee et al., 2013*) and NhaP (*Paulino et al., 2014*; *Wöhlert et al., 2014*) also form dimers, similar structures are observed at their dimer interfaces (*Figure 3—figure supplement 1b–c*). The conformations of T1 and T6 are stabilized by extensive and mostly hydrophobic interactions with the neighboring T2 and T7, respectively (*Figure 2b*). In CitS, NapA and NhaP, a number of neighboring structural elements contribute to the stabilization of the drastically different conformations of the T1/T6 equivalents (*Figure 3—figure supplement 1a–e*).

Our structure also revealed extensive interactions between the β and γ subunits. Each γ subunit interacts with one β subunit (*Figure 3—figure supplement 1f*). The interface buries 3100 Å$^2$ of surface area and is primarily composed of residues with hydrophobic side chains. T6 in the β subunit makes a significant contribution to the interface. However, if it is located away from the core domain like its equivalent in CitS, it is not expected to interact with the γ subunit. Additional interactions are mediated by T4 and T10.

The structural analysis suggested that positions of T1 and T6 are critical for the β subunit trimerization and binding the γ subunit. To probe the functional roles of the positions of T1 and T6, we introduced substitutions in the β subunit to promote them adopting conformations similar to their equivalents in CitS. T1, T6 and neighboring regions were substituted with equivalent regions in *Se*CitS. Consistent with our hypothesis, we found that the substituted β/CitS protein no longer

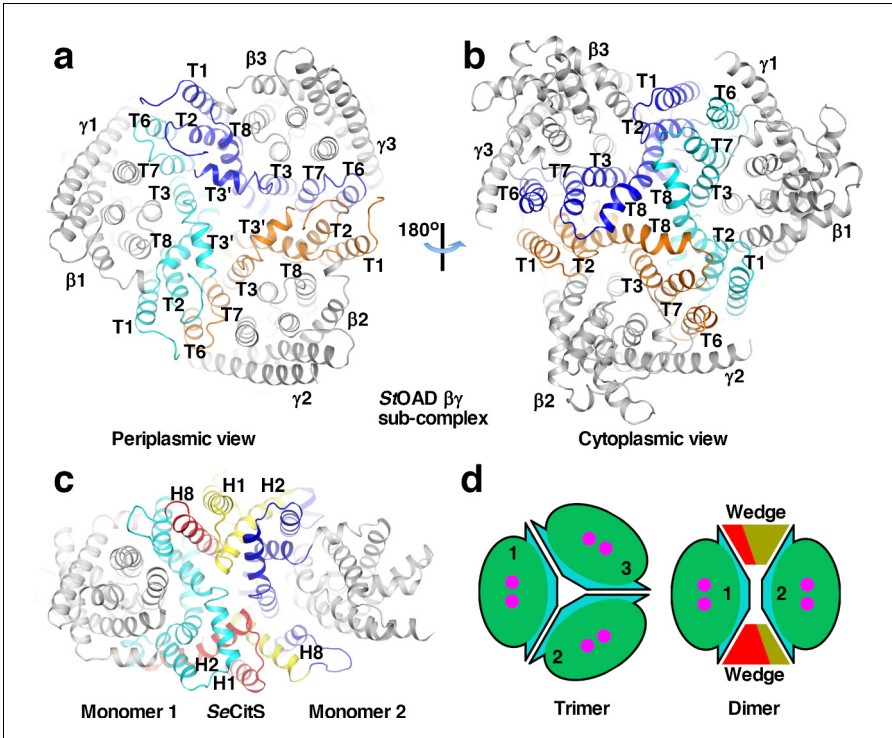

**Figure 3.** Interactions between *St*OAD β subunits. (**a**)-(**b**) Structure of the *St*OAD βγ sub-complex. The scaffold domains in the three β subunits are highlighted in cyan, orange and blue. Their secondary structure elements are labeled. The β subunit core domains and the γ subunits are colored in gray. (**c**) Structure of the *Se*CitS dimer (PDB 5A1S). Monomer 1 in the inward-open conformation is aligned to β1 in the *St*OAD βγ sub-complex in panel (**a**). Monomer 2 is in the outward-open conformation. Structural elements constituting the 'wedge' in monomers 1 and 2 are labelled and highlighted in red and yellow, respectively. Additional elements in the scaffold domain are colored in cyan and blue, respectively. (**d**) Schematic drawing of the *St*OAD β subunit trimer and the *Se*CitS dimer. The monomers are colored as in *Figure 1a*.

The online version of this article includes the following figure supplement(s) for figure 3:

**Figure supplement 1.** Conformational differences between T1 and T6 and their equivalents in the β subunit homologues.

interacts with the γ subunit (*Figure 3—figure supplement 1g*), and is likely dimeric (*Figure 3—figure supplement 1h*).

## Sodium binding sites in the β subunit

Several cavities are found on in the *St*OAD βγ sub-complex. A large cavity opening to the periplasm is found at the center of the β subunit trimer. The densities in it were modeled as three n-dodecyl-D-maltoside (DDM) molecules used in the complex purification (*Figure 4a*). This cavity is primarily formed by residues with hydrophobic side chains in T3', T3 and T8, suggesting that it may not contribute to sodium binding. In addition, each β subunit contains two negatively charged cavities in the cytoplasmic face. The first cavity is formed between the core and scaffold domains, about 8 Å deep and measures 10 Å by 8 Å at the opening. The second adjacent to it is much shallower (*Figure 1d*). The first cavity is highly conserved among decarboxylase sodium pumps, whereas residues in the second are more variable (*Figure 1—figure supplement 5c–d*). This suggests that the first cavity may be a binding site for the positively charged sodium ion, which should exist in all decarboxylase sodium pumps.

Additional hints for sodium binding sites come from structural studies of the β subunit homologues. Sodium or sodium-mimicking ions have been observed in structures of CitS (*Kim et al., 2017*; *Wöhlert et al., 2015*), NhaP (*Wöhlert et al., 2014*) and ASBT (*Hu et al., 2011*). A sodium binding site is located between the tips of helical hairpins in CitS and at the equivalent positions in

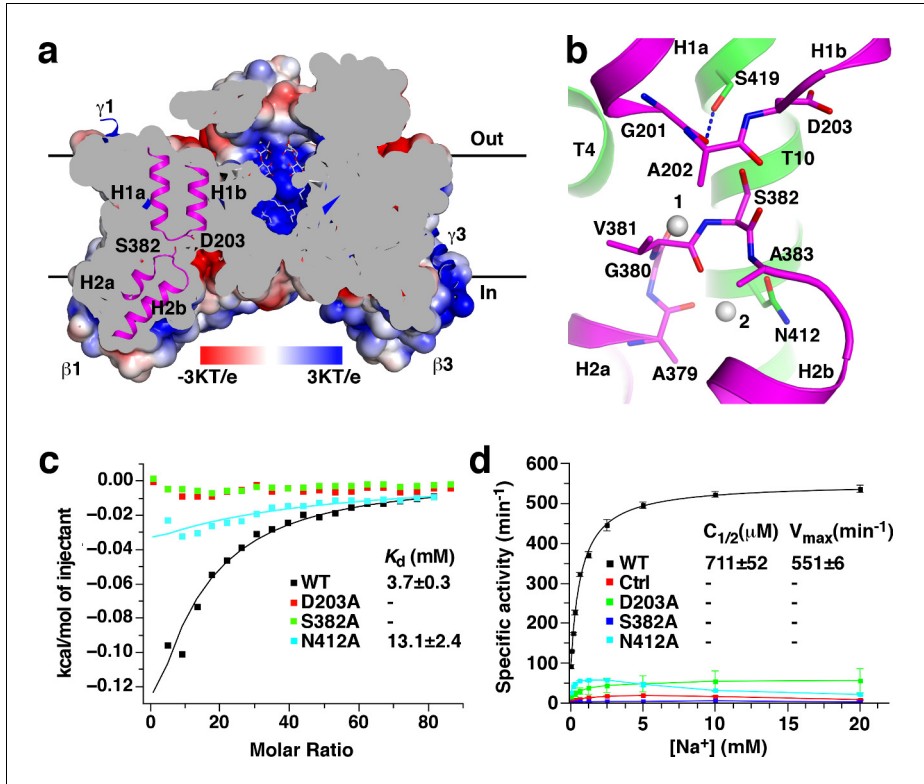

**Figure 4.** Putative sodium binding sites in the *St*OAD β subunit. (**a**) Cutaway view of the solvent-accessible surface of the *St*OAD β₃ sub-complex. The surface is colored according to the electrostatic potential. Helical hairpins H1, H2 and the Asp203 and Ser382 side chains at their tips are highlighted. The γ subunits are presented in cartoon representation and colored in blue. The DDM molecules in the cavity at the center of the complex are presented in stick representation with their carbon atoms in gray. (**b**) Structure of the putative sodium binding sites in the β subunit. The gray spheres 1 and 2 indicate equivalent positions of the first and second sodium binding sites in CitS, respectively. Polar or charged groups in the proximity are highlighted. (**c**) ITC experiments probing sodium binding to the wild type *St*OAD βγ sub-complex and variants with substitutions at the putative β subunit sodium binding sites. Errors are derived from fitting the data to the one set of sites binding model. (**d**) Oxaloacetate decarboxylation by the wild type *St*OAD and variants with substitutions at the putative β subunit sodium binding sites. The activity was reconstituted by supplementing the α subunit with detergent-solubilized βγ sub-complex. The specific activity is defined as the reaction velocity divided by the amount of *St*OAD. In the control experiment (Ctrl), the βγ sub-complex was omitted in the reaction mixture. The error bars represent standard deviations of three independent experiments. $C_{1/2}$, sodium concentration at half-maximum activity; $V_{max}$, maximum specific activity. Values of $C_{1/2}$ and $V_{max}$ and their errors were derived from data fitting.

The online version of this article includes the following source data and figure supplement(s) for figure 4:

**Source data 1.** Summary of ITC experiments.

**Source data 2.** Summary of OAD activity experiments.

**Figure supplement 1.** Thermograms of ITC experiments probing sodium binding to the wild type and substituted *St*OAD βγ sub-complex.

NhaP and ASBT. In the β subunit, the highly conserved H1 and H2 tips meet at the bottom of the putative sodium binding cavity (*Figure 4a*). The H1 and H2 tips each contain one residue with polar or charged side chains, Asp203 and Ser382, respectively. Their side chains point into the space between the helical hairpin tips (*Figure 4b*). Such structure suggests that the Asp203 and Ser382 side chains may participate in sodium binding.

To probe the functions of Asp203 and Ser382 in sodium binding, we introduced the D203A and S382A substitutions. Gel filtration experiments indicated that neither substitution affected the overall structure of the *St*OAD βγ sub-complex (*Figure 1—figure supplement 2*). We next determined their effects on sodium binding by isothermal titration calorimetry (ITC). The ITC experiments

revealed a $K_d$ of 3.7 mM for sodium binding to the wild type *St*OAD βγ sub-complex. Remarkably, the D203A and S382A substitutions completely abolished sodium binding (*Figure 4c*, *Figure 4—figure supplement 1* and *Figure 4—source data 1*), indicating that the Asp203 and Ser382 side chains make critical contributions to sodium binding. In line with the previous reports that continuous oxaloacetate decarboxylation by OAD requires sodium (*Dimroth, 1982*; *Dimroth and Thomer, 1988*) and a functional β subunit (*Di Berardino and Dimroth, 1996*; *Jockel et al., 2000a*; *Jockel et al., 2000b*), our activity assays indicated that these substitutions also severely inhibited oxaloacetate decarboxylation by *St*OAD. Whereas detergent-solubilized wild type *St*OAD βγ sub-complex strongly stimulated oxaloacetate decarboxylation by the α subunit in the presence of sodium, such activity of the substituted *St*OAD βγ sub-complexes was lost (*Figure 4d* and *Figure 4—source data 2*).

In *Se*CitS, a second sodium ion is bound near the tip of its helical hairpin 2. It is coordinated by polar groups in helical hairpin two and the nearby Ser427 side chain (*Wöhlert et al., 2015*). The residue equivalent to Ser427 in the *St*OAD β subunit is the highly conserved Asn412 (*Figure 4b*). To probe the function of the Asn412 side chain in sodium binding, we introduced the N412A substitution, which did not affect the overall structure of the *St*OAD βγ sub-complex (*Figure 1—figure supplement 2*). We found that this substitution caused more than 60% reduction in the heat generated in ITC experiments and a 3.5-fold increase in $K_d$ (*Figure 4c*, *Figure 4—figure supplement 1* and *Figure 4—source data 1*), which suggests a significant change in sodium binding thermodynamics. It also severely inhibited oxaloacetate decarboxylation by *St*OAD (*Figure 4d* and *Figure 4—source data 2*). Together, these data indicate an important role of the Asn412 side chain in sodium binding.

OAD transports two sodium ions per reaction cycle (*Dimroth and Thomer, 1993*) and its β subunit likely contains two sodium binding sites (*Jockel et al., 2000b*). The spatial locations of the Asp203, Ser382 and Asn412 side chains (*Figure 4b*) suggest that Asp203 and Ser382 contribute to the first sodium binding site, and Asn412 contributes to the second. In *Se*CitS, sodium ions bind to the second site only when the first is occupied (*Wöhlert et al., 2015*). Our data showed that the D203A and S382A substitutions severely inhibited sodium binding and the N412A substitution had a milder effect, suggesting that the two sodium binding sites in the OAD β subunit are also occupied sequentially, consistent with the reported cooperative binding of two sodium ions to OAD (*Jockel et al., 2000b*).

In our structure, the putative sodium binding sites are accessible from the cytoplasm but not the periplasm (*Figure 4a*). This indicates that the structure captured the β subunit in the inward-open conformation.

## Mutagenesis screen for essential residues in the β subunit

In many biotin-dependent enzymes, polar or charged amino acid side chains play critical roles in their carboxyl-transfer reactions in which biotin serves as the carboxyl carrier (*Tong, 2013*; *Waldrop et al., 2012*). To obtain insights into carboxyl-biotin decarboxylation by the β subunit, we surveyed β subunit residues with such side chains at the cytoplasmic face, where the carboxyl-biotin decarboxylation likely take place. We focused on highly conserved residues as functionally important residues are usually conserved. A total of 17 such residues were found (*Figure 5a* and *Figure 1—figure supplement 5d*). Among them, the equivalents of Thr148, Asp149, Tyr227, Thr318, Arg389, Asn392 and Ser419 have been characterized in the *Klebsiella pneumonia* OAD (*Kp*OAD) (*Di Berardino and Dimroth, 1996*; *Jockel et al., 2000a*; *Jockel et al., 2000b*). It was found that their substitutions did not severely inhibit oxaloacetate decarboxylation (*Supplementary file 1B*). In addition, side chains of Arg250, Glu298 and Asn315 are minimally exposed to solvent (*Figure 5a*) and may not play significant roles in carboxyl-biotin decarboxylation. After excluding these residues, as well as Asp203, Ser382 and Asn412 that we have characterized in the previous section, we probed the functions of Glu40, Arg258, Arg304 and Asn311 with mutagenesis.

We introduced the Glu40, Arg258, Arg304 and Asn311 to alanine substitutions. None of the substitutions were found to affect the overall structure of the *St*OAD βγ sub-complex (*Figure 1—figure supplement 2*). We next accessed their effects on oxaloacetate decarboxylation (*Figure 5b* and *Figure 5—source data 1*) and sodium binding (*Figure 5c*, *Figure 5—figure supplement 1* and *Figure 5—source data 2*). A severe inhibition of oxaloacetate decarboxylation was observed for the E40A substitution. This substitution also moderately reduced the affinity to sodium and caused a

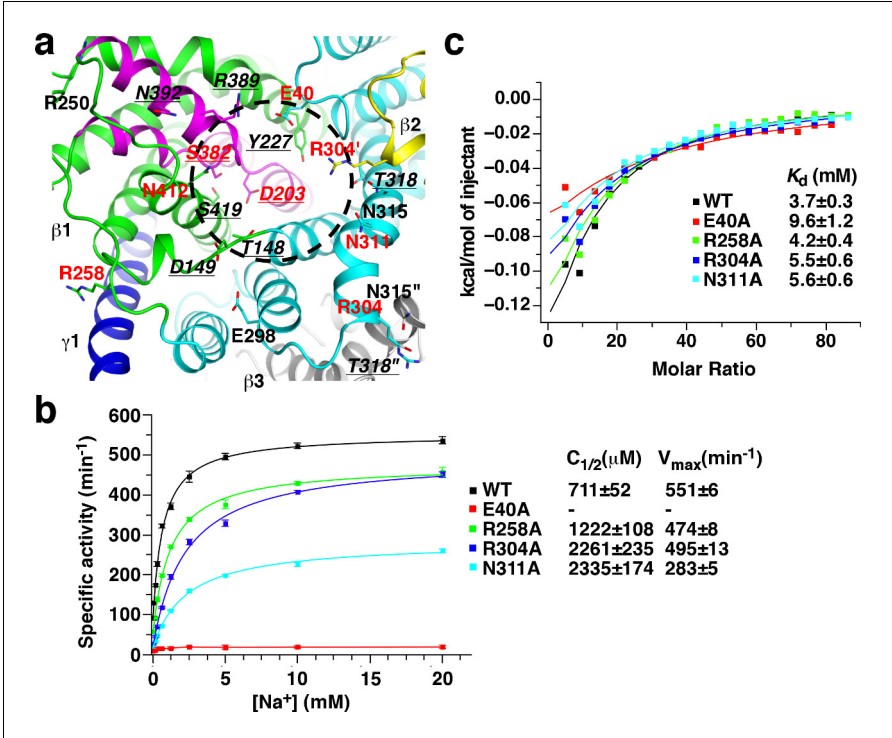

**Figure 5.** Mutagenesis screen for residues essential for the function of the β subunit. (**a**) Structure of the β subunit cytoplasmic face. Highly conserved residues with polar or charged side chains are highlighted. Residues with red and underlined labels were characterized by our study and previous studies on *Kp*OAD, respectively. The ′ and ″ signs indicate residues in the second (yellow) and third (gray) β subunits in the complex, respectively. The black cycle indicates the putative sodium binding cavity. (**b**) Oxaloacetate decarboxylation by *St*OAD with substitutions in the β subunit cytoplasmic face. Data for the wild type complex is shown for comparison. (**c**) ITC experiments probing sodium binding to *St*OAD βγ sub-complex with substitutions in the β subunit cytoplasmic face. Data for the wild type complex is shown for comparison.

The online version of this article includes the following source data and figure supplement(s) for figure 5:

**Source data 1.** Summary of OAD activity experiments.

**Source data 2.** Summary of ITC experiments.

**Figure supplement 1.** Thermograms of ITC experiments probing sodium binding to the *St*OAD βγ sub-complex.

---

2.6-fold increase in $K_d$. The R258A substitution had little effects on either oxaloacetate decarboxylation or sodium binding. The R304A and N311A substitutions moderately inhibited oxaloacetate decarboxylation. Both substitutions caused a 3-fold increase in the sodium concentration for half-maximum activity ($C_{1/2}$); the N311A substitution also reduced the maximum activity more than 40%. Neither of them had strong effects on sodium binding. Together, these data suggest that the Glu40 side chain is critical for the carboxyl-biotin decarboxylation by the β subunit but does not make significant contributions to its sodium binding, the Arg304 and Asn311 side chains are also important, the Arg258 side chain is unlikely to play a major role.

## Effect of pH on sodium binding to the *St*OAD βγ sub-complex

During the OAD catalysis, the β subunit acquires a proton from the periplasm and donates it to carboxyl-biotin decarboxylation (*Di Berardino and Dimroth, 1996*; *Dimroth and Thomer, 1983*). To test if proton and sodium binding to the β subunit affects each other, we repeated our ITC experiments at pH ranging from 5.55 to 8.45, corresponding to nearly 1000-fold change in proton concentration. We found that sodium binding is significantly affected by pH (*Figure 6—figure supplement 1a–b* and *Figure 6—source data 1*). Decreasing the pH from 8.45 to 5.55 increased the $K_d$ more than 2-fold (*Supplementary file 1C*). The $K_d$ value is largely unaffected by different choices of buffer (*Supplementary file 1C*), suggesting that the observed effect is primarily due to differences in

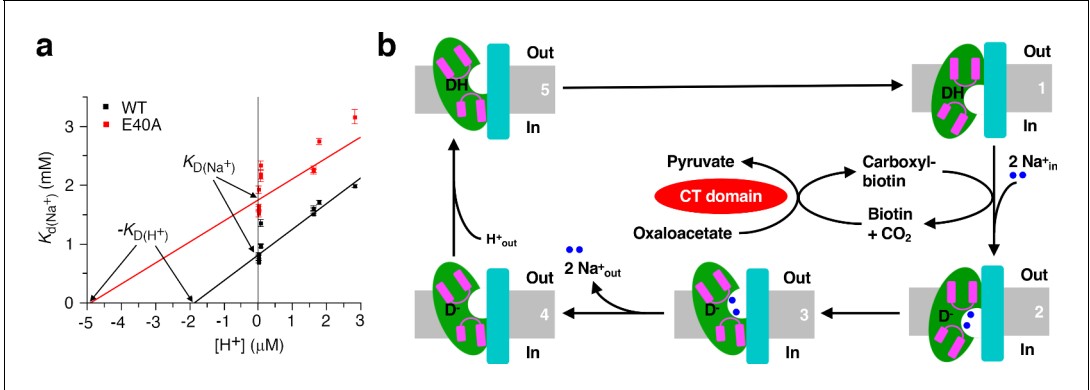

**Figure 6.** Insights into the OAD reaction cycle. (a) Sodium binding to the *St*OAD βγ sub-complex is affected by pH. The dissociation constants ($K_{d(Na+)}$) derived from ITC experiments at different pH is plotted against the proton concentration. Error bars represent errors derived from fitting the ITC data to the one set of sites model. The solid lines indicate data fitting with the competitive binding model. The y- and x-axis intercepts of the lines indicate the values for the absolute dissociation constants for sodium and proton ($K_{D(Na+)}$ and $K_{D(H+)}$), respectively. (b) A model of the OAD-catalyzed sodium transport. Components in the β subunit are colored as in *Figure 1*. For simplicity, only one β subunit is shown and its domain E is omitted. The gray bars indicate the membrane bilayer. The letter 'D' indicates the direct proton donor for carboxyl-biotin decarboxylation.

The online version of this article includes the following source data and figure supplement(s) for figure 6:

**Source data 1.** Summary of ITC experiments at different pH.

**Source data 2.** Dissociation constants of sodium binding to the *St*OAD βγ sub-complex at different pH measured by ITC.

**Source data 3.** Enthalpy of sodium binding to the *St*OAD βγ sub-complex at different pH measured by ITC.

**Figure supplement 1.** ITC experiments probing sodium binding to the wild type (a) and the E40A substituted (c) *St*OAD βγ sub-complex at different pH.

**Figure supplement 2.** ITC-measured enthalpy of sodium binding to the *St*OAD βγ sub-complex does not correlate with the buffer protonation enthalpy.

**Figure supplement 3.** OAD β subunit residues characterized by previous mutagenesis studies.

**Figure supplement 4.** model of the OAD holoenzyme.

proton concentration. The $K_d$ value increases linearly with increasing proton concentration (*Figure 6a* and *Figure 6—source data 2*), consistent with competitive binding of proton and sodium (*Leone et al., 2015*). Our structural and binding data suggest that sodium ions bind to a cavity in the cytoplasmic face of the inward-open β subunit (*Figure 4a–c*). Our cryo-EM and crystallographic studies performed at pH 7.5 and 5.5, respectively, captured the β subunit in the inward-open conformation, suggesting that it is the preferred conformation in our experimental conditions over a large pH range. Together, these data suggest that proton and sodium compete for binding to same cavity in the inward-open β subunit. Absolute dissociation constants for binding in the absence of competitors can be derived from fitting the data to the competitive binding model (*Hariharan and Guan, 2017*; *Leone et al., 2015*). Our data indicated that the absolute dissociation constants for sodium ($K_{D(Na+)}$) and proton ($K_{D(H+)}$) are 0.81 ± 0.03 mM and 1.9 ± 0.2 μM, respectively (*Figure 6a*). The p$K_a$ for the β subunit proton-binding group calculated from the logarithm of $K_{D(H+)}$ is 5.73. A similar effect is observed for the E40A substituted *St*OAD βγ sub-complex (*Figure 6a*, *Figure 6—figure supplement 1c*, *Figure 6—source data 1* and *Supplementary file 1C*).

In a number of previous studies to characterize the sodium/proton competition for membrane protein binding, ITC experiments performed at the same pH revealed a strong correlation between the sodium binding enthalpy and buffer protonation enthalpy (*Hariharan and Guan, 2017*; *Leone et al., 2015*). In these studies, sodium releases proton from the protein through competition. The subsequent proton binding to the buffer molecules contributes to the observed correlation. We also tested a number of buffers with different protonation enthalpies (*Supplementary file 1C*; *Goldberg et al., 2002*), but did not observe such correlation (*Figure 6—figure supplement 2a–b* and *Figure 6—source data 3*). The estimated p$K_a$ for the β subunit proton-binding group suggests that it is close to 50% protonated at pH 5.79, at which pH a group of our ITC experiments were performed. The lack of correlation suggests that a large fraction of the released proton do not

bind to the buffer molecules. A close inspection of the *St*OAD β subunit structure revealed a cluster of four solvent-exposed histidine residues at the periplasmic face (*Figure 6—figure supplement 2c*). These histidine residues could provide binding sites for the released proton in our ITC experiments.

## Discussion

The putative sodium binding sites we proposed are supported by previous mutagenesis studies on *Kp*OAD. Consistent with their role in coordinating sodium, it was found that substitutions on the Asp203 and Ser382 equivalents in *Kp*OAD (Asp203 and Ser382) severely disrupted its function (*Di Berardino and Dimroth, 1996*; *Jockel et al., 2000b*). In addition, our structure indicated that the Ser419 side chain forms a hydrogen bond with the Gly201 main chain carbonyl in the H1 tip (*Figure 4b*), suggesting that Ser419 plays a role in stabilizing the first sodium binding site. Consistent with such a role, it was found that substituting the Ser419 equivalent in *Kp*OAD (Ser419) with alanine reduced its oxaloacetate decarboxylation activity by 90% (*Jockel et al., 2000a*).

Our mutagenesis screen identified a number of residues essential for the β subunit's function, including Glu40, Asp203, Arg304, Asn311, Ser382 and Asn412 (*Figure 5a*). Our data suggests that the Asp203, Ser382 and Asn412 side chains coordinate the sodium ions required for the OAD catalysis. They and side chains of Glu40, Arg304 and Asn311, may have additional functions. For instance, serval of them may participate in coordinating the biotin moiety, and/or in relaying the proton required for carboxyl-biotin decarboxylation. These residues are all located in the putative sodium-binding cavity (*Figure 5a*). Our ITC experiments suggest that sodium and proton compete for binding to this cavity (*Figure 6a*). Together, these data are consistent with a model that carboxyl-biotin decarboxylation takes place in the sodium-binding cavity. Sodium binding to the cavity could facilitate the release of proton for carboxyl-biotin decarboxylation, providing an explanation for the sodium requirement for continuous oxaloacetate decarboxylation by OAD (*Dimroth, 1982*). However, it remains to be established that the proton competes with sodium for binding is the same proton for carboxyl-biotin decarboxylation, and the possibility that this reaction takes place elsewhere cannot be ruled out. To pinpoint the location of carboxyl-biotin decarboxylation and understand the functional roles of the important residues we identified a clear picture of how carboxyl-biotin interacts with the β subunit is required.

Both inward- and outward-open conformations have been captured by structural studies on several sodium transporters including CitS (*Kim et al., 2017*; *Wöhlert et al., 2015*), NapA (*Coincon et al., 2016*; *Lee et al., 2013*) and NhaP (*Paulino et al., 2014*; *Wöhlert et al., 2014*). These studies revealed that an 'elevator-like' motion of their substrate binding domain against their scaffold domain transports the substrates across the membrane (*Drew and Boudker, 2016*). Their substrate binding and scaffold domains are homologous to the core and scaffold domains in the *St*OAD β subunit, respectively. In the *St*OAD β subunit, the interface between the core and scaffold domains are primarily composed of hydrophobic residues, which could facilitate a similar elevator-like motion. In addition, our data suggests that sodium ions probably bind to equivalent locations in the *St*OAD β subunit, CitS and NhaP. These data suggest that the OAD β subunit may also utilize a similar 'elevator mechanism' for sodium transport (*Figure 6b*). The competition of sodium and proton for binding to the inward-open β subunit (*Figure 6a*) is consistent with a model that the 'elevator' movement also transport the proton for carboxyl-biotin decarboxylation in the opposite direction (*Figure 6b*). Our model is consistent with the reported reversibility of decarboxylase sodium pumps (*Dimroth and Hilpert, 1984*), and may provide a mechanism for the observed inside/outside sodium exchange by OAD in the absence of pyruvate and oxaloacetate (*Dimroth and Thomer, 1993*). In this condition the enzyme cannot catalyze functional cycles, either forward or backward; however, the β subunit can interconvert between sodium-bound outward- and inward-open conformations, and thus catalyze the passive exchange of sodium across the membrane.

Our ITC experiments and proposed OAD reaction cycle suggest a precise coordination of the OAD components during the reaction. Under most physiological conditions, the sodium concentration exceeds the proton concentration a million- to a billion- fold. The absolute dissociation constants for proton and sodium determined by our ITC experiments differ less than 500-fold. Therefore, under most physiological conditions, the inward-open β subunit is prone to lose the bound proton due to competitive sodium binding. If it is the same proton for carboxyl-biotin decarboxylation, our model suggests that the inward proton transport by the β subunit (state 5 to 1

depicted in *Figure 6b*) is accompanied by a synchronized carboxyl-biotin transfer to the β subunit, or the proton is lost to solution by competitive sodium binding.

Based on conservation and predicted locations in the protein, a number of residues in the *Kp*OAD β subunit have been selected for mutagenesis studies (*Di Berardino and Dimroth, 1996*; *Jockel et al., 2000a*; *Jockel et al., 2000b*). Combining the mutagenesis data and our structure provided insights into their function (*Figure 6—figure supplement 3a* and *Supplementary file 1B*). In addition to Asp203 and Ser382, it was found that substitutions on Tyr229, Asn373 and Gly377 in the *Kp*OAD β subunit also severely affected its function (*Jockel et al., 2000a*; *Jockel et al., 2000b*). The equivalent residues in the *St*OAD β subunit are Tyr229, Asn373 and Gly377. The Tyr229 side chain in T5 forms a hydrogen bond with the Leu343 main chain carbonyl in the nearby T9 (*Figure 6—figure supplement 3b*). The Asn373 side chain N-terminal to H2 forms a hydrogen bond with the Glu249 main chain carbonyl C-terminal to T5 (*Figure 6—figure supplement 3c*). Gly377 in H2 packs against T9 and the space between them cannot accommodate an amino acid side chain (*Figure 6—figure supplement 3d*). Therefore, Tyr229, Asn373 and Gly377 are important for the proper folding of the core domain. Substitutions on these residues likely disrupt the function of the β subunit by distorting its structure.

The extensive interactions between the β and γ subunits suggest that they form a similar $\beta_3\gamma_3$ hetero-hexamer in the OAD holoenzyme. In a recent study on *Vc*OAD2, blue native PAGE analysis of the complex revealed multiple species; two species with molecular weights of 530 kDa and 750 kDa were found to contain all three subunits (*Balsera et al., 2011*). These two species may represent βγ sub-complexes associated with different numbers of α subunits, suggesting that the association of the α subunit and the βγ sub-complex is dynamic. Our blue native PAGE experiments of the anti-HA agarose-purified *St*OAD complex also revealed species of similar molecular weights (*Figure 3—figure supplement 1h*), although it remains to be seen whether they contain the α subunit. The γ subunit C-terminal tail binds to the α subunit with a 1:2 molar ratio (*Balsera et al., 2011*), suggesting that the $\beta_3\gamma_3$ hetero-hexamer can bind up to 6 α subunits. The α subunit consists of CT, AD and BCCP domains. The isolated α subunit is dimeric (*Balsera et al., 2011*), probably through dimeric interactions of its CT domain (*Studer et al., 2007*). In our structure, the last residue visible in the three γ subunits (Phe43) are located ~65 Å apart. A simple modeling indicates that three α subunit dimers can simultaneously interact with the three γ subunit C-terminal tails in the $\beta_3\gamma_3$ hetero-hexamer (*Figure 6—figure supplement 4*). The modelled α subunits dimerize through their CT domains. One of the AD domains in each α subunit dimer interacts with the γ subunit C-terminal tail. The model suggests that interactions between the α subunit dimers may also exist and play a role in stabilizing the OAD complex, consistent with a recent study that the isolated γ subunit C-terminal tail promotes the formation of α subunit tetramers (*Balsera et al., 2011*). It was suggested by that study that the α subunit tetramerization is mediated by the AD domains in different α subunit dimers. In the γ subunit, the predicted amphipathic helix that likely interacts with the AD domain (*Balsera et al., 2011*) is linked to the transmembrane helix with a 20-residues linker (*Figure 1—figure supplement 1b*). This linker could provide flexibility and allow the AD domains in different α subunit dimers to interact (*Figure 6—figure supplement 4*).

The OAD holoenzyme could provide a platform to facilitate potential coordination of its components. For instance, the γ subunit interacts with both the core and scaffold domains in the β subunit (*Figure 1a*). In the proposed 'elevator' movement for sodium transport, the relative orientation of the core and scaffold domains changes drastically (*Figure 6b*). A synchronous conformational change in the γ subunit is expected, which may be in turn coupled to conformational changes in the α subunit. Such synchronization of the α and β subunits may play a role in coordinating their activities.

Our study provided important insights into OAD's function. However, a full understanding of the molecular mechanism requires extensive future studies. The 'elevator mechanism' we proposed is largely based on structural homologies to CitS and other transporters, which needs to be verified and improved by additional structures of the β subunit at different stages in the reaction cycle. Structural studies of the β subunit in complex with biotin and other substrates are needed to elucidate where carboxyl-biotin decarboxylation takes place and how it drives the sodium transport. Importantly, how oxaloacetate decarboxylation is coupled to sodium transport is poorly understood. Structural studies of the OAD holo-enzymes could provide valuable insights into this process.

# Materials and methods

## Key resources table

| Reagent type (species) or resource | Designation | Source or reference | Identifiers | Additional information |
|---|---|---|---|---|
| Gene (*Salmonella typhimurium*) | *St*OAD | GenBank | GenBank: CP007235.2, bases 836524–839869 | |
| Strain, strain background (*Escherichia coli*) | BL21 star (DE3) | Thermo Fischer | C602003 | Chemical competent cells |
| Antibody | anti-HA (rabbit monoclonal) | Cell Signaling Technology | Cat# 3724S | 1:1000 |
| Antibody | HRP-linked anti-rabbit IgG (goat polyclonal) | Cell Signaling Technology | Cat# 7074P2 | 1:2000 |
| Antibody | HRP-linked anti-6x Histidine tag (goat polyclonal) | Abcam | Cat# ab1269 | 1:5000 |
| Recombinant DNA reagent | *St*OAD-pET28a (plasmid) | This paper, Materials and methods | | *St*OAD expersion, can be obtained from the Xiang lab |
| Recombinant DNA reagent | *St*OAD-β/CitS-pET28a (plasmid) | This paper, Materials and methods | | *St*OAD β/CitS variant expersion, can be obtained from the Xiang lab |
| Recombinant DNA reagent | *St*OADα-pET28a (plasmid) | This paper, Materials and methods | | *St*OAD α subunit expersion, can be obtained from the Xiang lab |
| Commercial assay or kit | NativePAGE Novex Bis-Tris gel | Thermo Fisher | Cat# BN1001BOX | |
| Software, algorithm | DYNAMICS V6 | Wyatt Technologies | | |
| Software, algorithm | MotionCor2 | *Zheng et al., 2017* | | |
| Software, algorithm | CTFFIND4 | *Rohou and Grigorieff, 2015* | | |
| Software, algorithm | e2boxer.py | *Tang et al., 2007* | | |
| Software, algorithm | RELION | *Scheres, 2012* | | |
| Software, algorithm | EMAN | *Ludtke et al., 1999* | | |
| Software, algorithm | EMBUILDER | *Zhou et al., 2017* | | |
| Software, algorithm | COOT | *Emsley and Cowtan, 2004* | | |
| Software, algorithm | PHENIX | *Adams et al., 2010* | | |
| Software, algorithm | ResMap | *Kucukelbir et al., 2014* | | |
| Software, algorithm | Dali server | *Holm and Rosenström, 2010* | | |
| Software, algorithm | HKL2000 | *Otwinowski and Minor, 1997* | | |
| Software, algorithm | MOLREP | *Vagin and Teplyakov, 1997* | | |
| Software, algorithm | ORIGIN 7.0 | Originlab | | |
| Software, algorithm | QTIPLOT | www.qtiplot.com | | |

## Protein expression and purification

The *St*OAD operon was amplified from the *Salmonella typhimurium* genome and inserted into vector pET28a (Novagene). The resulting recombinant protein contains a 6x Histidine tag at the N-terminus of the γ subunit. For protein expression, this plasmid and a pTYB12 (New England Biolabs)-based plasmid containing the *E. coli* biotin ligase *birA* gene were co-transformed into the *E. coli* strain BL21 star (DE3). The cells were cultured in Luria-Bertani broth supplemented with 50 mg/liter

kanamycin and 100 mg/liter ampicillin, induced with 0.5 mM isopropyl-β-D-thiogalactopyranoside (IPTG, Bio Basic) at 16℃ for 16 hr. 15 mg/liter of D-biotin was supplemented prior to induction. Harvested cells were lysed in a buffer containing 20 mM Tris pH 7.5, 300 mM sodium chloride, 2 mM β-mercaptoethanol (β-ME) and 0.2 mM phenylmethylsulfonyl fluoride (PMSF) with an AH-2010 homogenizer (ATS Engineering). The lysate was cleared by centrifugation at 6,000 g for 15 min, and membrane fractions were collected by additional centrifugation at 200,000 g for 1 hr. Membrane protein were extracted by incubating the membrane fraction with a buffer containing 20 mM Tris pH 7.5, 300 mM sodium chloride, 2 mM β-ME, 0.2 mM PMSF and 1% DDM for 12 hr at 4℃. After clearing the suspension by ultracentrifugation at 200,000 g for 30 min, extracted *St*OAD was purified with Ni-NTA agarose (Qiagen) and size exclusion (Superose 6 increase 10/300 GL, GE Healthcare) columns, concentrated to 10 mg/ml in a buffer containing 20 mM Tris pH 7.5, 100 mM sodium chloride, 2 mM dithiothreitol (DTT) and 0.02% DDM, flash cooled in liquid nitrogen and stored at −80℃. A gel shift assay indicated that the α subunit in the purified *St*OAD complex was fully biotinylated.

To purify the *St*OAD βγ sub-complex, the sodium chloride concentration in the Ni-NTA-purified *St*OAD was reduced to 100 mM and the solution was loaded to a Hitrap Q column (GE Healthcare). Pure *St*OAD βγ sub-complex was collected from the flow-through, and further purified by Hitrap Heparin (GE Healthcare) and size exclusion (Superose 6 increase 10/300 GL, GE Healthcare) columns. The purified complex was concentrated to 10 mg/ml in a buffer containing 20 mM Tris pH 7.5, 100 mM sodium chloride, 2 mM DTT and 0.02% DDM, flash cooled in liquid nitrogen and stored at −80℃.

To purify the *St*OAD α subunit from cells expressing the *St*OAD complex, the soluble fraction of the cell lysate was incubated with monomeric avidin agarose (Pierce). Bound protein was eluted with the lysis buffer supplemented with 2 mM D-biotin.

To purify the *St*OAD α subunit for activity assays, the corresponding gene fragment was inserted into vector pET28a. The plasmid was co-transformed with the *birA*-containing plasmid into the *E. coli* strain BL21 star (DE3). After supplementing 15 mg/liter of D-biotin, the cells were induced with 0.5 mM IPTG at 16℃ for 16 hr. Fully biotinylated 6x Histidine tagged *St*OAD α subunit was purified with Ni-NTA and size exclusion (Superdex 200 increase 10/300 GL) columns, concentrated to 10 mg/ml in a buffer containing 100 mM potassium phosphate (pH 6.5) and 0.02% DDM, flash cooled in liquid nitrogen and stored at −80℃.

To construct the expression plasmid for the *St*OAD β/CitS variant, gene fragments for the *Se*CitS residues 1–79 and 238–319 were codon optimized for expression in *E. coli* and synthesized (Genewiz). These fragments were used to replace gene fragments for *St*OAD β subunit residues 1–39 and 247–300 in the *St*OAD expression construct, respectively. A HA-coding oligonucleotide was added to the 5′ end of the coding region for the β/CitS variant or the wild type β subunit. To purify the HA-tagged *St*OAD or the β/CitS variant, membrane fraction of cell lysates from cells expressing the corresponding complexes were incubated with anti-HA agarose (Pierce), and bound protein was eluted with a buffer containing 20 mM Tris (pH 7.5), 100 mM sodium chloride, 0.02% DDM and 2 mg/ml HA peptide (Sangon Biotech).

Amino acid substitutions were generated with the QuikChange kit (Agilent Technologies) and verified by DNA sequencing. The substituted complexes were expressed and purified following the same protocol for the wild type complex.

## Blue native PAGE

Blue native PAGE was performed with the precast NativePAGE Novex Bis-Tris gel (4–16%, Thermo Fisher Scientific) following the manufacturer's directions. All samples were exchanged into a buffer containing 50 mM Hepes (pH 7.4), 20 mM sodium chloride and 1% DDM before loaded to the gel. NativeMark unstained protein standard (Thermo Fisher Scientific) was used as the molecular weight marker.

## Antibodies

The anti-HA Rabbit mAb (3724S, Cell Signaling Technology) and horseradish peroxidase (HRP)-linked anti-rabbit IgG (7074P2, Cell Signaling Technology) anti-bodies were used to detect the HA tag. The HRP-linked anti-6x Histidine tag antibody (ab1269, Abcam) was used to detect the 6x Histidine tag.

## Dynamic light scattering

Dynamic light scattering experiments were performed on a DynaPro NanoStar instrument (Wyatt Technologies) at 25°C. The StOAD βγ sub-complex was characterized at a concentration of 36 μM, in a buffer containing 20 mM Tris pH7.5, 100 mM sodium chloride, 2 mM DTT and 0.02% DDM. Data was analyzed with the DYNAMICS V6 software (Wyatt Technologies).

## Structure determination and analysis

To prepare grids for cryo-EM studies, 3 μl of StOAD βγ sub-complex were applied to a glow-discharged 400-mesh holy carbon grid (Quantifoil R1.2/1.3, Cu). After blotting away excess sample by filter paper for 3.5 s in an automatic plunge device (Leica EMGP), the grids were flash plunged into liquid ethane. Cryo-EM data was collected on a Titan Krios G2 transmission electron microscope equipped with a direct detector with GIF quantum energy filter (K2 summit). The grids were imaged at super resolution mode with a magnification of 105,000, yielding a pixel size of 0.68 Å. The energy filtered mode was used with a 20 eV slit inserted. A total dose of 50 electrons/$Å^2$ from 9.25 s exposure was dose fractionated into 32 frames. The defocus of collected micrographs ranges from 1 μm to 3.5 μm. The beam-induced motion was corrected by MotionCor2 (*Zheng et al., 2017*). The defocus and astigmatism parameters were estimated by CTFFIND4 (*Rohou and Grigorieff, 2015*). 56,052 particles were picked by e2boxer.py (*Tang et al., 2007*) from a sub-dataset. The featured 2D class averages produced by RELION (*Scheres, 2012*) 2D classification were used as references for automatic particle picking by RELION, which picked 260,982 particles from 821 selected micrographs (without dose weighting). After two rounds of reference-free 2D classification, six good classes with 129,525 particles were selected for further data processing. The initial model was built by StartCSym in EMAN (*Ludtke et al., 1999*) with C3 symmetry imposed. In the 3D classification without imposing symmetry, only one good class with 75,699 particles were selected for 3D refinement. These particles were re-extracted from dose-weighted micrographs and were refined with C3 symmetry imposed. The resolution of the final reconstruction is 3.88 Å, determined by gold standard Fourier Shell Correlation (FSC) at cut-off 0.143 with post-process in RELION. An initial structure of the StOAD βγ sub-complex was built with EMBUILDER (*Zhou et al., 2017*). The structure was completed with COOT (*Emsley and Cowtan, 2004*) and refined with real space refinement implemented in PHENIX (*Adams et al., 2010*). Local resolution was calculated with ResMap (*Kucukelbir et al., 2014*). Structural homologues were identified with the Dali server (*Holm and Rosenström, 2010*).

The purified StOAD complex crystallized with vapor diffusion. The reservoir solution contains 100 mM sodium chloride, 100 mM lithium sulfate, 100 mM sodium citrate (pH 5.5), 25% polyethylene glycol (PEG)1000 and 0.125% PEG400. SDS PAGE analysis of the crystals indicated that they do not contain the α subunit. A diffraction data set to a resolution of 4.4 Å was collected at the Shanghai Synchrotron Radiation Facility Beamline BL17U1, at the wavelength 0.979 Å, and processed with the HKL2000 package (*Otwinowski and Minor, 1997*). The crystals belong to space group C222$_1$ and contain 3 β and 3 γ subunits in the asymmetric unit. The crystal structure was determined with molecular replacement with MOLREP (*Vagin and Teplyakov, 1997*), using the cryo-EM structure as the search model. Structural refinement was carried out with PHENIX.

## Isothermal titration calorimetry

ITC experiments were performed on an ITC 200 instrument (MicroCal) at 25°C. To access the effects of the β subunit substitutions on sodium binding, the StOAD βγ sub-complex was exchanged into a solution containing 100 mM potassium phosphate (pH 6.5) and 0.02% DDM. To access the effects of pH and buffer on sodium binding, the StOAD βγ sub-complex was exchanged into a solution containing 20 mM of the specified buffer, 100 mM potassium chloride and 0.02% DDM. The buffers used in this study were Mes pH 5.55, Mes pH 5.75, Tris pH 8.45, Citrate pH 5.79, Mes pH 5.79, Bis-Tris pH 5.79, Pipes pH 7.2, Hepes pH 7.2, Tris pH 7.2, Hepes pH 8.2, Tricine pH 8.2 and Tris pH 8.2. Hydrochloric acid and tetramethylammonium hydroxide were used to adjust the pH. To characterize sodium binding, the StOAD βγ sub-complex at a concentration of 120 μM was transferred to a 200 μl cell. A solution with 50 mM sodium chloride in the matching buffer was injected into the cell, 2 μl at a time.

ITC data were analyzed with ORIGIN 7.0 (Originlab). The OAD β subunit contains two sodium binding sites (*Jockel et al., 2000b*), which are probably not identical judging from our structure

(*Figure 4b*). However, the thermograms of our ITC experiments appear to be uniphasic, making data fitting with more than one set of binding sites difficult (*Brautigam, 2015*). We fitted the data to the one set of sites model, with the stoichiometry number fixed to 2. In our ITC experiments the Wiseman '*c*' parameter is significantly smaller than one and it is very difficult to increase it due to difficulties in concentrating the *St*OAD βγ sub-complex sample. In this situation, fixing the stoichiometry number to a known value can restrain data fitting and obtain more accurate results (*Turnbull and Daranas, 2003*).

### Oxaloacetate decarboxylation by *St*OAD

Oxaloacetate decarboxylation was monitored by following the absorption change at 265 nm due to oxaloacetate (extinction coefficient: 0.95 $mM^{-1}cm^{-1}$) (*Dimroth, 1986*) on an ultraspec 2100 pro spectrophotometer (GE Healthcare). The light path length is 1 cm. The *St*OAD βγ sub-complex were exchanged into a buffer containing 100 mM potassium phosphate (pH 6.5) and 0.02% DDM before the experiment. The reaction mixture contains 300 mM Tris pH 7.5, 2 mM oxaloacetate, 0.79 µM 6x Histidine tagged *St*OAD α subunit, 5 µM *St*OAD βγ sub-complex and sodium chloride at indicated concentrations. Data analysis was performed with QTIPLOT (www.qtiplot.com). The specific activity is defined as reaction velocity divided by the amount of *St*OAD. To determine the values for sodium concentration at half maximum activity ($C_{1/2}$) and the maximum specific activity ($V_{max}$), data were fitted to the following equation:

$$V = \frac{(V_{max} - V_0)\left( C + C_{OAD} + C_{1/2} - \sqrt{\left( C + C_{OAD} + C_{1/2} \right)^2 - 4CC_{OAD}} \right)}{2C_{OAD}} + V_0$$

Additional symbols in the equation stand for specific activity (V), specific activity at zero sodium concentration ($V_0$), sodium concentration (C), *St*OAD concentration ($C_{OAD}$). This equation was deduced by assuming that *St*OAD saturated by sodium has a maximum specific activity $V_{max}$, and sodium binds to *St*OAD with a $K_d$ that equals to $C_{1/2}$.

## Acknowledgements

We thank the Shanghai Synchrotron Radiation Facility Beamline BL17U1 for setting up the beamline and assistance during diffraction data collection, the Center for Biological Imaging, Core Facilities for Protein Science at the Institute of Biophysics, Chinese Academy of Sciences and Xiaojun Huang, Boling Zhu, Gang Ji, Prof. Fei Sun and other staff members for support during cryo-EM data collection, Prof. Peng Zhang at Institute for Plant Physiology and Ecology, Chinese Academy of Sciences for assistance in membrane protein preparation, the Core Facility for Molecular Biology at Institute of Biochemistry and Cell Biology, Chinese Academy of Sciences for assistance with dynamic light scattering and ITC experiments. This work is supported by Natural Science Foundation of China (general grants 31870769 and 31570743 to SX).

## Additional information

### Funding

| Funder | Grant reference number | Author |
| --- | --- | --- |
| National Natural Science Foundation of China | 31870769 | Song Xiang |
| National Natural Science Foundation of China | 31570743 | Song Xiang |

The funders had no role in study design, data collection and interpretation, or the decision to submit the work for publication.

### Author contributions

Xin Xu, Huigang Shi, Xiaowen Gong, Pu Chen, Investigation; Ying Gao, Xinzheng Zhang, Supervision, Investigation, Project administration; Song Xiang, Conceptualization, Data curation, Formal analysis,

Supervision, Funding acquisition, Validation, Investigation, Visualization, Methodology, Project administration

## Author ORCIDs

Song Xiang https://orcid.org/0000-0001-9314-4684

## Decision letter and Author response

Decision letter https://doi.org/10.7554/eLife.53853.sa1
Author response https://doi.org/10.7554/eLife.53853.sa2

## Additional files

### Supplementary files

• Supplementary file 1. Supplementary data. (**A**) Crystal diffraction data collection and structure refinement statistics.(**B**) Previous mutagenesis studies performed on KpOAD. (**C**) Summary of ITC experiments at different pH.

• Transparent reporting form

### Data availability

The cryo-EM structure of the StOAD βγ sub-complex and related data have been deposited into the protein data bank (https://www.pdb.org) and the electron microscopy data bank (https://www.ebi.ac.uk/pdbe/emdb/), with the accession numbers 6IWW and EMD-9743, respectively. The crystal structure of the StOAD βγ sub-complex and the diffraction data have been deposited into the protein data bank with the accession number 6IVA. Source data for Figure 4c-d, Figure 4-figure supplement 1, Figure 5b-c, Figure 5-figure supplement 1, Figure 6a, Figure 6-figure supplement 1, and Figure 6-figure supplement 2a-b are provided.

The following datasets were generated:

| Author(s) | Year | Dataset title | Dataset URL | Database and Identifier |
|---|---|---|---|---|
| Xu X, Shi H, Zhang X, Xiang S | 2018 | Cryo-EM structure of the S. typhimurium oxaloacetate decarboxylase betagamma sub-complex | https://www.ebi.ac.uk/pdbe/entry/emdb/EMD-9743 | Electron Microscopy Data Bank, EMD-9743 |
| Xu X, Shi H, Zhang X, Xiang S | 2018 | Crystal structure of the S. typhimurium oxaloacetate decarboxylase betagamma sub-complex | http://www.rcsb.org/structure/6IVA | RCSB Protein Data Bank, 6IVA |

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
