## [Decision Letter]

**Acceptance summary:**

We are excited to publish this study as we believe it showcases a fascinating a membrane protein system whereby a proton-dependent enzymatic reaction and a sodium-proton antiport mechanism become structurally coupled to produce a more complex biological function – one that evolved to be crucial to the bioenergetics of anaerobic microorganisms.

**Decision letter after peer review:**

Thank you for submitting your article "Structural Insights into Sodium Transport by the Oxaloacetate Decarboxylase Sodium Pump" for consideration by *eLife*. Your article has been reviewed by three peer reviewers, and the evaluation has been overseen by José D. Faraldo-Gómez as Reviewing Editor and Kenton Swartz as the Senior Editor. Reviewer #2 has agreed to reveal her identity (Lan Guan). Reviewers #1 and #3 have opted to remain anonymous.

The reviewers have discussed the reviews with one another and the Reviewing Editor has drafted this decision to help you prepare a revised submission.

Summary:

Many microorganisms utilize transmembrane electrochemical gradients of sodium ions, rather than protons, to power the recycling of ADP into ATP among other essential metabolic and membrane transport processes. Here, Xiang and co-workers report a partial structure of a bacterial protein complex whose function is to generate such a Na^+^ gradient, through a conformational cycle that couples a decarboxylation reaction with Na^+^ translocation across the membrane. Specifically, the manuscript describes a cryo-EM structure for the β and γ submits of the oxaloacetate decarboxylase (OAD) complex. The OAD complex decarboxylates oxaloacetate to pyruvate. This chemical process somehow results in sodium extrusion across the membrane. In this way the catabolism of oxaloacetate is used to power the generation of a sodium gradient. To date, the molecular mechanism of OAD has remained enigmatic. The manuscript describes an important piece of the puzzle. Interestingly, the structure of the β subunit is highly similar to other secondary-active transporters belonging to the CPA/AT superfamily. More specifically, the structure of OAD most closely resembles the sodium-coupled citrate symporter (CitS). The structure of CitS has been determined in multiple conformations and has been shown to operate by an elevator alternating-access mechanism. Based on this structural similarity, a mechanism for OAD is proposed. Combined with functional and binding assays, this study constitutes a significant advancement in our understanding of this class of systems, and will no doubt foster new experimental and theoretical work. The study was perceived by reviewers as technically sound, scientifically rigorous, and presented logically. These merits notwithstanding, reviewers and editors agree that key aspects of the study require further development before the manuscript can be accepted for publication in *eLife*.

Essential revisions:

1) It is essential for the authors to show the mutants used in the functional assay are well folded (gel filtration traces).

2) The mechanistic model outlined in Figure 6 describes two very different – and seemingly contradictory – processes. When the transporter is outward-facing, the release of Na^+^ permits the recognition of an external proton; binding of Na^+^ and H^+^ to this state is therefore implicitly assumed to be competitive, as in conventional sodium-proton antiporters. The model postulates something very different for the inward-facing conformation, however – namely that Na^+^ must bind concurrently with H^+^, and that only upon decarboxylation of carboxyl-biotin is the H^+^ released (i.e. transferred to biotin). If confirmed, the notion that Na^+^ and H^+^ binding to the same transporter can be synergistic in one conformation but competitive in another would be quite remarkable. Thus, we believe this element requires further analysis. As has been demonstrated for other membrane protein systems that bind Na^+^ and H^+^ (Leone et al., 2015; Hariharan and Guan, 2017), ITC experiments carefully designed to minimize initial Na^+^ contamination and carried out at different pH values and/or with different buffers of varying protonation enthalpy can convincingly probe whether binding of Na^+^ and H^+^ is synergistic or competitive, and in some cases, reveal their stoichiometric ratio. It seems likely that the purified protein sample examined by the authors by ITC primary reflects the same conformation observed in the cryo-EM images, i.e. the inward-facing state of the β subunit. The authors should therefore pursue this type of experiments, particularly for the WT and if possible also for selected mutants (e.g. Glu40), and contrast their results with the hypotheses outlined in Figure 6. Use of Tris-HCl buffer may help this assay if Na^+^ and H^+^ binding are interdependent.

3) The ITC results should include the thermograms, which can be shown as supplementary information. Also, please clarify what stoichiometry number for Na^+^ was used for fitting (in all cases), and what justifies this choice?

4) The authors appear to gloss over the fact that the β subunit of the OAD was thought to resemble other CPA transporters as highlighted in the transporter classification data-base (http://www.tcdb.org/search/result.php?tc=3.B.1) and Wikipedia for example (https://en.wikipedia.org/wiki/Sodium transporting_carboxylic_acid_decarboxylase). The structure of the β subunit of OAD is interesting, but the major question is how decarboxylation of oxaloacetate to pyruvate is coupled to Na^+^ transport? This manuscript does not appear to answer this question, as the α subunit of OAD could not be co-purified for structural determination. The authors should make it much clearer what "has" and "has not" been shown by this study. In reality, the authors have presented an elevator alternating-access mechanism that is inferred from that observed the citrate transporter CitS and more distantly related Na+/H? exchangers (*eLife* 2015 4:e09375; Scientific Reports 2017 7:2548). The authors should make this point clearer as the proposed mechanism in Figure 6 seems to be largely based on structural homologues of OAD β subunit captured previously in different conformations.

[Editors' note: further revisions were suggested prior to acceptance, as described below.]

Thank you for resubmitting your article "Structural Insights into Sodium Transport by the Oxaloacetate Decarboxylase Sodium Pump" for consideration by *eLife*. Your revised article has been reviewed by the three experts who evaluated the original version, and the evaluation has been overseen by Kenton Swartz as Senior Editor. The reviewers have opted to remain anonymous.

The reviewers have discussed the reviews with one another and the Reviewing Editor has drafted this decision to help you prepare a revised submission.

Summary:

Many microorganisms utilize transmembrane electrochemical gradients of sodium ions, rather than protons, to power the recycling of ADP into ATP among other essential metabolic and membrane transport processes. Here, Xiang and co-workers report a partial structure of a bacterial protein complex whose function is to generate such a Na^+^ gradient, through a conformational cycle that couples a decarboxylation reaction with Na^+^ translocation across the membrane. Specifically, the manuscript describes a cryo-EM structure for the β and γ submits of the oxaloacetate decarboxylase (OAD) complex. The OAD complex decarboxylates oxaloacetate to pyruvate. This chemical process somehow results in sodium extrusion across the membrane. In this way the catabolism of oxaloacetate is used to power the generation of a sodium gradient. To date, the molecular mechanism of OAD has remained enigmatic. The manuscript describes an important piece of the puzzle. Interestingly, the structure of the β subunit is highly similar to other secondary-active transporters belonging to the CPA/AT superfamily. More specifically, the structure of OAD most closely resembles the sodium-coupled citrate symporter (CitS). The structure of CitS has been determined in multiple conformations and has been shown to operate by an elevator alternating-access mechanism. Based on this structural similarity, a mechanism for OAD is proposed, whereby the catalytic reaction powers the translocation of Na^+^ uphill, utilizing a H^+^ in the process. This system is thus distinct from primary-active ion transporters, which are energized by ATP hydrolysis, and from most secondary-active transporters, which are energized by downhill uptake of ions. Combined with functional and binding assays, this study constitutes a significant advancement in our understanding of this class of systems, and will no doubt foster new experimental and theoretical work. The study was perceived by reviewers as technically sound and scientifically rigorous. These merits notwithstanding, reviewers and editors agreed that key aspects of the study required further development, and invited a revision. The reviewers found the revised manuscript to have improved. However, key concerns remain pertaining to the interpretation and discussion of some of the results. No new experimental data is required at this point, but the authors are strongly encouraged to pursue experiments designed to reveal the location of the site of decarboxylation reaction.

Essential revisions:

1) The newly provided ITC data probing the interplay between Na^+^ and H^+^, most probably for the inward facing state, is a strong indication that these ions compete for the same binding site, as in the conventional Na+/H^+^ antiporters that are closely related to this system. However, neither the mechanistic diagram presented in Figure 6 or the related text seem to have been altered in the light of this key finding. The competitive model that stems from the ITC data implies that state 2 does not actually exist as drawn, i.e. an inward-facing state with both Na^+^ and H^+^ bound concurrently. Instead, the new data suggests that the state that follows state 1 must be one in which the proton has already transferred out (either to the decarboxylation site or other proton-buffering site in the protein or the solvent) before Na^+^ ions can bind. This would be in analogy to the authors' own description of the subsequent steps in the cycle, for the outward-facing states, i.e. Na^+^ comes off between states 5 and 4 before a proton binds between states 5 and 6. The authors must correct their scheme and explanation so that they are consistent with the newly provided ITC data – which was requested precisely to challenge the notion that competition occurs in the outward state but not in the inward state.

2) In globular proteins (e.g., *E. coli* biotin carboxylase) carboxyl-biotin decarboxylation reactions seem to require extensive coordination of biotin in a large domain interface, and to be further mediated via several metal ions (Waldrop et al., 2012). In the absence of direct evidence of a bound substrate, the claim that the enzymatic reaction occurs in the sodium binding cavity (Figure 5) seems too speculative – and should be discussed as such, outlining alternative interpretations and possible approaches to clarify this question. For example, it would appear just as plausible for the reaction to take place in the oligomerization interface, somehow. It is worth noting that *E. coli* biotin carboxylase forms a complex with ATP in the absence of biotin. By analogy future experiments could assay and identify nucleotide binding sites in OAD.

3) We strongly encourage the authors to trim down their manuscript to focus on the main scientific contributions and to improve its readability. Possible sections where the text could be more concise include the Introduction, e.g. the preamble regarding topology discrepancies or discussion of other members in the OAD family; likewise, in Results, it is unclear what mechanistic insights are gained through the comparison of the scaffold domains in β OAD to CitS and NapA and the different oligomerization of these proteins; this section could be more concise, or omitted entirely – with details added to the supplementary figures captions if needed. Finally, in Discussion, there is some repetition of the structural analysis presented earlier.

---

## [Author Response]

Essential revisions:1) It is essential for the authors to show the mutants used in the functional assay are well folded (gel filtration traces).

We have added Figure 1—figure supplement 2 to show the gel filtration traces. The mutants behave similarly as the wild type complex on the gel filtration column, suggesting that they are well folded.

2) The mechanistic model outlined in Figure 6 describes two very different – and seemingly contradictory – processes. When the transporter is outward-facing, the release of Na^+^ permits the recognition of an external proton; binding of Na^+^ and H^+^ to this state is therefore implicitly assumed to be competitive, as in conventional sodium-proton antiporters. The model postulates something very different for the inward-facing conformation, however – namely that Na^+^ must bind concurrently with H^+^, and that only upon decarboxylation of carboxyl-biotin is the H^+^ released (i.e. transferred to biotin). If confirmed, the notion that Na^+^ and H^+^ binding to the same transporter can be synergistic in one conformation but competitive in another would be quite remarkable. Thus, we believe this element requires further analysis. As has been demonstrated for other membrane protein systems that bind Na^+^ and H^+^ (Leone et al., 2015; Hariharan and Guan, 2017), ITC experiments carefully designed to minimize initial Na^+^ contamination and carried out at different pH values and/or with different buffers of varying protonation enthalpy can convincingly probe whether binding of Na^+^ and H^+^ is synergistic or competitive, and in some cases, reveal their stoichiometric ratio. It seems likely that the purified protein sample examined by the authors by ITC primary reflects the same conformation observed in the cryo-EM images, i.e. the inward-facing state of the β subunit. The authors should therefore pursue this type of experiments, particularly for the WT and if possible also for selected mutants (e.g. Glu40), and contrast their results with the hypotheses outlined in Figure 6. Use of Tris-HCl buffer may help this assay if Na^+^ and H^+^ binding are interdependent.

We have performed the suggested ITC experiments with the wild type and the E40A substituted *St*OAD βγ sub-complex, which indicated that increasing pH increased the affinity between sodium and the βγ sub-complex. We have added our crystallographic study of the *St*OAD βγ sub-complex to the manuscript, which also captured the β subunit in the inward-open conformation. Our crystallographic and cryo-EM studies were performed at pH 5.5 and 7.5, respectively, suggesting that the inward-open conformation is preferred in our experimental conditions over a large pH range. Together, the ITC and structural data suggest that sodium and proton compete for binding to the negatively charged pocket in the inward-open β subunit. In the studies referenced by the reviewers(Hariharan and Guan, 2017; Leone, Pogoryelov, Meier, and Faraldo-Gomez, 2015), a strong correlation between the sodium binding enthalpy and the buffer protonation enthalpy is found. In these studies, proton is released from the protein by the competitive sodium binding. Its subsequent binding to buffer molecules contributes to the correlation. Such correlation allows the estimation of the sodium/proton stoichiometric ratio. In our experiments, such correlation was not observed, suggesting that a large fraction of the released proton does not bind to the buffer molecules. The β subunit contains a solvent exposed cluster of four histidine residues, which may bind to the released proton in our ITC experiments. We have added a “Effect of pH on Sodium binding to the *St*OAD βγ sub-complex” section to describe these experiments. We have also revised the Discussion to discuss the insights these new experiments brought into the mechanism of OAD.

3) The ITC results should include the thermograms, which can be shown as supplementary information. Also, please clarify what stoichiometry number for Na^+^ was used for fitting (in all cases), and what justifies this choice?

We have added Figure 4—figure supplement 1 and Figure 5—figure supplement 1 to show the thermograms. It has been reported that the OAD β subunit contains two sodium binding sites (Jockel, Schmid, Steuber, and Dimroth, 2000) and our structure indicates that they are probably not identical (Figure 4B). However, the thermograms of our ITC experiments appear to be uniphasic, making data fitting using more than one set of binding sites difficult(Brautigam, 2015). We fitted the ITC data with the one set of sites model. In our ITC experiments, the concentration of the *St*OAD βγ sub-complex (120 mM) is significantly smaller than *K*_d_ (in the millimolar range) and cannot be significantly increased due to difficulties in concentrating the sample. This makes the Wiseman “*c*” parameter less than 1, in which case fixing the stoichiometry number to a known number can restrain data fitting to obtain more accurate results (Turnbull and Daranas, 2003). We fixed the stoichiometry number to 2 since the β subunit contains two sodium binding sites. We have revised the related paragraph in the Materials and methods section to include the above discussion.

4) The authors appear to gloss over the fact that the β subunit of the OAD was thought to resemble other CPA transporters as highlighted in the transporter classification data-base (http://www.tcdb.org/search/result.php?tc=3.B.1) and Wikipedia for example (https://en.wikipedia.org/wiki/Sodium transporting_carboxylic_acid_decarboxylase). The structure of the β subunit of OAD is interesting, but the major question is how decarboxylation of oxaloacetate to pyruvate is coupled to Na^+^ transport? This manuscript does not appear to answer this question, as the α subunit of OAD could not be co-purified for structural determination. The authors should make it much clearer what "has" and "has not" been shown by this study. In reality, the authors have presented an elevator alternating-access mechanism that is inferred from that observed the citrate transporter CitS and more distantly related Na+/H? exchangers (eLife 2015 4:e09375; Scientific Reports 2017 7:2548). The authors should make this point clearer as the proposed mechanism in Figure 6 seems to be largely based on structural homologues of OAD β subunit captured previously in different conformations.

We have revised the third paragraph in the “Structure of the *St*OAD β subunit” section to reference previous studies that suggested a homology between the OAD β subunit and members in the CPA2 family. We agree with the reviewers that our current study did not answer many important questions regarding OAD’s function including how oxaloacetate decarboxylation is coupled to sodium transport, and that the “elevator mechanism” is primarily based structural homology to other transporters and needs to be validated and improved. We have added a last paragraph in the Discussion section to discuss the limitations of the current study and what future studies are needed. We thought that future structural studies of the β subunit at different stages of the reaction cycle and on the OAD holo-enzyme are needed to fully understand the molecular mechanism of OAD.

[Editors' note: further revisions were suggested prior to acceptance, as described below.]

Essential revisions:1) The newly provided ITC data probing the interplay between Na^+^ and H^+^, most probably for the inward facing state, is a strong indication that these ions compete for the same binding site, as in the conventional Na+/H^+^ antiporters that are closely related to this system. However, neither the mechanistic diagram presented in Figure 6 or the related text seem to have been altered in the light of this key finding. The competitive model that stems from the ITC data implies that state 2 does not actually exist as drawn, i.e. an inward-facing state with both Na^+^ and H^+^ bound concurrently. Instead, the new data suggests that the state that follows state 1 must be one in which the proton has already transferred out (either to the decarboxylation site or other proton-buffering site in the protein or the solvent) before Na^+^ ions can bind. This would be in analogy to the authors' own description of the subsequent steps in the cycle, for the outward-facing states, i.e. Na^+^ comes off between states 5 and 4 before a proton binds between states 5 and 6. The authors must correct their scheme and explanation so that they are consistent with the newly provided ITC data – which was requested precisely to challenge the notion that competition occurs in the outward state but not in the inward state.

We have revised Figure 6B and paragraphs two and three in the Discussion section to better interpret the new ITC data. We have removed state 2 from Figure 6B, which is inconsistent with our new ITC data. The competitive binding of sodium and proton suggests that sodium may facilitate the proton release for carboxyl-biotin decarboxylation, we have included this discussion in paragraph two in the Discussion section.

2) In globular proteins (e.g., *E. coli* biotin carboxylase) carboxyl-biotin decarboxylation reactions seem to require extensive coordination of biotin in a large domain interface, and to be further mediated via several metal ions (Waldrop et al. 2012). In the absence of direct evidence of a bound substrate, the claim that the enzymatic reaction occurs in the sodium binding cavity (Figure 5) seems too speculative – and should be discussed as such, outlining alternative interpretations and possible approaches to clarify this question. For example, it would appear just as plausible for the reaction to take place in the oligomerization interface, somehow. It is worth noting that *E. coli* biotin carboxylase forms a complex with ATP in the absence of biotin. By analogy future experiments could assay and identify nucleotide binding sites in OAD.

We agree that our hypothesis is speculative and other possibilities cannot be ruled out. We have revised paragraph one in the “Effect of pH on Sodium binding to the *St*OAD βγ sub-complex” section and paragraph two in the Discussion section to make it clear. We have also revised the last paragraph in the Discussion section to point out that structural studies of the b subunit in complex with substrates required for carboxyl-biotin decarboxylation are needed to pinpoint the location of this reaction and understand its mechanism. Biotin carboxylase couples ATP hydrolysis with the carboxyl transfer from bicarbonate to biotin. However, neither biotin carboxylation nor carboxyl-biotin decarboxylation by OAD requires ATP or other nucleotides. We are not aware of any reports that nucleotides are involved in the OAD reaction.

3) We strongly encourage the authors to trim down their manuscript to focus on the main scientific contributions and to improve its readability. Possible sections where the text could be more concise include the Introduction, e.g. the preamble regarding topology discrepancies or discussion of other members in the OAD family; likewise, in Results, it is unclear what mechanistic insights are gained through the comparison of the scaffold domains in β OAD to CitS and NapA and the different oligomerization of these proteins; this section could be more concise, or omitted entirely – with details added to the supplementary figures captions if needed. Finally, in Discussion, there is some repetition of the structural analysis presented earlier.

We appreciate this suggestion and have trimmed down the manuscript at the suggested places.